# Adhesion Behaviour of Primary Human Osteoblasts and Fibroblasts on Polyether Ether Ketone Compared with Titanium under In Vitro Lipopolysaccharide Incubation

**DOI:** 10.3390/ma12172739

**Published:** 2019-08-27

**Authors:** Korbinian Benz, Andreas Schöbel, Marisa Dietz, Peter Maurer, Jochen Jackowski

**Affiliations:** 1Department of Oral Surgery and Dental Emergency Care, Faculty of Health, Witten/Herdecke University, 58455 Witten, Germany; 2Department of Oral and Maxillofacial Surgery, Hospital North Dortmund, 44145 Dortmund, Germany; 3Private Practice Clinic for Oral Surgery, 66606 St. Wendel, Germany

**Keywords:** human osteoblasts, human fibroblasts, titanium, PEEK, inflammation, cell adhesion

## Abstract

The aim of this in vitro pilot study was to analyse the adhesion behaviour of human osteoblasts and fibroblasts on polyether ether ketone (PEEK) when compared with titanium surfaces in an inflammatory environment under lipopolysaccharide (LPS) incubation. Scanning electron microscopy (SEM) images of primary human osteoblasts/fibroblasts on titanium/PEEK samples were created. The gene expression of the LPS-binding protein (LBP) and the LPS receptor (toll-like receptor 4; TLR4) was measured by real-time polymerase chain reaction (PCR). Immunocytochemistry was used to obtain evidence for the distribution of LBP/TLR4 at the protein level of the extra-cellular-matrix-binding protein vinculin and the actin cytoskeleton. SEM images revealed that the osteoblasts and fibroblasts on the PEEK surfaces had adhesion characteristics comparable to those of titanium. The osteoblasts contracted under LPS incubation and a significantly increased LBP gene expression were detected. This was discernible at the protein level on all the materials. Whereas no increase of TLR4 was detected with regard to mRNA concentrations, a considerable increase in the antibody reaction was detected on all the materials. As is the case with titanium, the colonisation of human osteoblasts and fibroblasts on PEEK samples is possible under pro-inflammatory environmental conditions and the cellular inflammation behaviour towards PEEK is lower than that of titanium.

## 1. Introduction

Three different material classes are available for implant systems: metallic materials (primarily titanium), ceramics, and polymers (plastics) such as polyether ether ketone (PEEK). With a share of 70%–80%, titanium and its alloys are the materials that are primarily used for implants [1]. The reasons for this are their excellent corrosion resistance and their high degree of mechanical stiffness and biocompatibility [2]. Titanium is, therefore, universally used, for example, in orthopaedic surgery and is routinely employed when teeth are replaced with dental implants [3,4]. Case reports of the positive function of oral implants extend back over 50 years, with the controlled reporting of clinical function dating back to around 25 years [5].

However, the use of titanium might have its limitations. Some studies have reported local inflammatory restrictions or even a loss of the implant after metal ions have been released [6,7]. Recent in vitro studies show the cytotoxic effect of titanium debris on human cells [8]. Moreover, titanium has an elasticity value (110 GPa), which is 5–10 times higher than that of human bones (approximately 10–20 GPa) [9]. The compression and tensile forces involved are transferred directly to the surrounding bones by the implant and result in von Mises comparative stress being applied to the bone–implant transition [10].

Recently published work showed an increased adhesion, viability, and proliferation of osteoblasts and gingival fibroblasts on PEEK and zirconia surfaces compared with titanium and supports the clinical relevance in this field [11].

Ceramic zirconium is an alternative to a purely metallic implant. Zirconium has a good biocompatibility, a low bacterial surface colonization, and suitable mechanical properties, but no long-term data of more than 10 years currently exist with regard to its clinical use [12].

Polyether ether ketone (PEEK) is a synthetically manufactured plastic that has a high degree of chemical resistance towards a wide range of organic and inorganic chemicals, acids, and biological fluids [13,14]. PEEK has an elasticity module of 3–4 gigapascal (GPa), which, although not identical with that of human cortical bones (10–20 GPa), is similar enough when compared to that of titanium implants [9,15]. The similar elasticity values of bone and PEEK result in any physiological burdens being distributed between the PEEK implant and the surrounding bones [16]. However, the reduced stiffness may also lead to higher stress peaks for this material. It has previously been successfully used in the field of prosthetic implants as a disk replacement material in orthopaedic surgery and for calvarial reconstructions because of its satisfactory chemical, thermal, and mechanical properties [17,18,19], as well as its biocompatibility [20].

Lipopolysaccharide (LPS) is the primary pathogenic factor of all Gram-negative bacteria and has been reported to trigger a strong inflammatory reaction [21]. The bacterial toxin LPS has a high binding affinity to biomaterials [22]. LPS exists as a polymeric aggregate when put in a watery environment [23]. Various lipids and serum proteins may connect with LPS [24]. The lipopolysaccharide binding protein (LBP) plays a key role among these proteins [25]. LBP is a 50 kilo Dalton (kDa) polypeptide. Before being released into the blood system as 58 to 60 kDa glycoprotein, it is primarily synthesized in the liver [26]. In addition to LPS, other lipopeptides can also be bound by the LBP protein [25]. Thus, LBP seems to recognize a multitude of microorganisms and plays a key role in the innate immune response to Gram-negative bacteria [27].

The binding of LBP to LPS causes the splitting of the LPS aggregate [28]. After binding to LBP, LPS seems to trigger a stronger inflammatory reaction in comparison with the dissolved form [29].

Osteoblasts are known to be able to detect the existence of bacteria by means of toll-like receptors (TLRs) [30]. The cellular expression and localisation of lipopolysaccharide (LPS)-binding protein (LBP) in human osteoblasts have not been described in the past.

For studying inflammatory responses, it is common to stimulate monocytes with lipopolysaccharide (LPS), a cell membrane component of Gram-negative bacteria [31]. Human monocytes have expressed significant levels of both TLR4 and TLR2 [32,33]. At constitutive concentrations, LBP is crucial for the host to sense bacteria. It also stimulates cells such as monocytes and macrophages to initiate an appropriate inflammatory response [34]. TLR4 activation in adventitial fibroblasts likely attracts monocytes through the production of inflammatory cytokines, which promotes fibroblast migration and proliferation [35].

The chronic inflammatory and foreign body response to orthopaedic implant byproducts has been well characterized [36]. Foreign materials and microorganisms are normally phagocytosed and degraded by neutrophils. As the size of implanted biomaterials usually is much larger than the leukocytes themselves, the standard degradation process is replaced by frustrated phagocytosis. Instead, leukocyte products like oxygen free radicals and lysosomal proteases are released to degrade the foreign body [37]. Within a day or two, the neutrophils are replaced by monocytes, lymphocytes, and macrophages, and the proliferation of connective tissue and blood vessels is initiated in the implant site [37]. Macrophages are also very important for the process of wound healing and the regeneration of the tissue [38].

As activated monocytes/macrophages induct the healing response at the implant site, fibroblasts and endothelial cells migrate into the granulation tissue [38]. Macrophages induced through the production of cytokines and growth factors the proliferation and migration of fibroblasts. Vice versa, fibroblasts can activate monocytes, suggesting synergistic interactions during matrix re-modeling [38]. Monocytes also play an important role in the development of new blood vessels during wound repair by differentiating into specific cell types as required by the injured tissue [39]. The growing of new blood vessels accelerates the migration of new cells. While activated monocytes and macrophages initiate the healing process, endothelial cells and fibroblasts start with the production of the first physiological tissue in form of collagen fibers around the implant [38].

After the implant has been inserted into the host tissue, its surface is first coated with plasma proteins from the blood and connective tissue. These first plasma proteins adhere minutes to hours after implantation and form a temporary extracellular matrix (ECM). It is commonly assumed that host cells interact with surface-adsorbed proteins rather than with the biomaterial surface itself [37]. The initial binding process of cells on bone implants may be mediated by the proteins adsorbed onto the biomaterial surface [37].

Some surfaces favour the adsorption of proteins such as albumin. Albumin has a “passivating” effect because albumin-coated material attracts macrophages less than uncoated material [37]. Other proteins can bind important anchor molecules for cellular contact, for example, fibronectin or vitronectin, and are thus conducive to colonisation by cells [40]. The extracellular matrix of cells consists of a complex mixture of matrix proteins. Some of these are glycoproteins such as fibronectin [41]. Fibronectin is a glycoprotein consisting of dimers with a molecular weight of about 250,000 Dalton (Da). Different kinds of fibronectin are synthesized by alternative splicing. An isoform of fibronectin, which circulates in soluble form within the blood serum, is produced in the liver [42]. Serum-contained fibronectin may apparently be bound to the surface of polymers and in turn be able to stimulate adhesion and adherence by activating receptors at the cell surface. Analysis of cell growth has shown that mitosis may strongly be increased by coating with fibronectin. Attachment molecules, such as laminin and fibronectin, attract cells to find first contact sites with an adhesion substrate’s surface and develop the adhesion of cells to ECM [42].

The titaniumoxid layer on the titanium surface supports the binding of proteins of the blood or cell culture medium. Among them is fibronectin, which supports the adhesion and osseo-integration of osteoblasts [43]. After cell adhesion and spreading, osteoblasts start producing bone-specific proteins like fibronectin [44].

Apart from the involvement in infectious diseases, TLR has been proposed to play a role in the development of autoimmune diseases, as some TLRs were shown to also recognize host-derived elements (e.g., TLR4 recognizes fibronectin and HSP60) [45].

Fibroblasts initially synthesize proteoglycans and fibronectin to create the matrix [46].

When blood gets in contact with an implanted material, a layer of host proteins adsorbs to the material surface right away; this includes blood proteins such as fibrinogen (Fg), fibronectin, and vitronectin [37]. It has also been described that fibronectin and vitronectin (both adhesion protheins of the ECM) attach to biomaterial surfaces [38]. Both are reported to be critical in regulating the inflammatory response compared with fibrinogen and complement. However, they also support the adhesion and spreading of osteoblasts to foreign materials, which plays the most important part in osseointegration of dental implants and points out the high effector capacity of the adsorbed protein layer [38].

Furthermore, gingival cells show reduced adhesion and spreading on collagen and fibronectin as compared with skin cells, suggesting that the repertoire and function of ECM receptors in gingival cells are distinct [47].

A pronounced denaturation of the proteins can be regarded as unfavourable for further interaction with the body, as denatured proteins are foreign to the body and may initiate inflammatory reactions.

Recent studies have shown good results for both zirconium and PEEK regarding early adhesion, proliferation, and the viability of osteoblasts and fibroblasts compared with titanium [11].

Pathological inflammatory processes that affect dental implants are defined as peri-implant disease. This may result in the loss of soft and hard tissue surrounding an osseointegrated implant [48]. To obtain a deeper understanding of the cellular reactions, the objective of this in vitro pilot study, the morphology and adhesion behaviour of human osteoblasts and fibroblasts on PEEK were analysed in comparison with titanium surfaces. Both cell types were also studied under LPS incubation, which simulated a bacterial infection that can occur in vivo, for example, after the insertion of a bone implant.

The working hypotheses were as follows:
When using PEEK, cell adhesion takes a place that is similar to that of titanium.The cellular inflammation behaviour towards PEEK is lower than that with titanium under LPS incubation.


## 2. Materials and Methods

### 2.1. Cell Seeding

Primary human osteoblasts (NHOst, Lot Number: 0000288136; LONZA, Basel, Switzerland) and fibroblasts (NHDF, Lot Number: 0061502; PromoCell, Heidelberg, Germany) were cultivated on titanium and PEEK samples (both: diameter: 12 mm, density: 2.5 mm; MEDICON, Tuttlingen, Germany). After shipping, the material probes were cleaned in 70% ethanol overnight to remove residue of the production by the companies PromoCell/LONZA. The next day, the samples were dried under a laminar floor and autoclaved (121 °C, 1 bar, 20 min.). The probes were stored at room temperature and, before use in the cell culture, all materials were sterilized by UV light for 15 min.

The cells were cultured following slight modifications of the protocols provided by the companies PromoCell and LONZA, respectively. The seeding density of the fibroblasts NHDF were 5000 cells/cm^2^ and for the NHOst, 11,000 osteoblasts/cm^2^. Instead of trypsin, Accutase was used for detaching the cells from the culture flasks. For the 24 h LPS incubation, LPS from the bacterium *Escherichia coli* (Sigma-Aldrich, Taufkirchen, Germany) was used at a concentration of 10 µg/mL, as provided by Tilakaratne et al. [49]. *E. coli* is able to bind to TLR4 and to trigger an inflammatory response. The handling of all human samples strictly adhered to the “Declaration of Helsinki”.

### 2.2. Scanning Electron Microscopy (SEM)

SEM images were created in order to analyse the morphology of the two cell types on the titanium and PEEK probes. Coverslips (Hecht Assistent, Sondheim, Germany) coated with poly-l-lysine protein (Sigma-Aldrich, Taufkirchen, Germany) were employed as the reference material. After the fixation of the cell samples, contrasting was carried out with 0.2% osmium tetroxide (Science Service, Düsseldorf, Germany). Subsequent treatment with hexamethyldisilazane (HMDS; Carl Roth, Karlsruhe, Germany) avoided the necessity of carrying out critical point drying. In order to improve the evaluation of the cell morphology, individual cells in the obtained images were manually coloured (Adobe Photoshop CS5; Adobe Systems, Munich, Germany).

### 2.3. Real-Time Polymerase Chain Reaction (PCR)

Real-time PCR was used to analyse the gene expression of the LPS-binding protein (LBP) and the LPS receptor (toll-like receptor 4; TLR4). The osteoblasts and fibroblasts were seeded on coverslips (coated with poly-l-lysine). The primers were obtained from Qiagen (Hilden, Germany). CyC1 (Cytochrome C) was the selected reference gene for the osteoblasts and Eif4A2 (eukaryotic initialisation factor 4A2) was selected for the fibroblasts, with both genes having been tested in preliminary studies. A kit from Qiagen (QuantiTect^®^ Reverse Transcription Kit; Hilden, Germany) was used for cDNA synthesis.

### 2.4. Immunocytochemical Marking

Evidence of the existence of LBP/TLR4 at the protein level and, additionally, of phalloidin (evidence of actin) and vinculin (extracellular matrix binding protein) was provided by immunocytochemical marking. The osteoblasts and fibroblasts were seeded in a density of 11,000 and 5000 cells/cm^2^ (24-well plate) on the materials coverslip, PEEK, and titanium (*n* = 8 probs per material) and cultivated for four days. After a further 24 h incubation with LPS (10 µg/mL) or only growth medium (each *n* = 4), the cover glasses and material samples were washed twice with PBS, followed by fixation of the cells with 4% paraformaldehyde solution (4% PFA in PBS). The next step was the blocking of endogenous peroxidases by 10% goat serum (normal goat serum, NGS; Life Technologies, Darmstadt, Germany) in PBS + 0.3% Triton X100 (Sigma-Aldrich, Taufkirchen, Germany) for 30 min at room temperature. The blocking solution also contained the first antibodies at a concentration of 1:75—rabbit anti-human LBP (PA5-21642, Thermo Scientific; Watham, MA, USA) and mouse anti-human TLR4 (76B357.1, (ab22048); Abcam, Cambridge, UK). The cover glasses and material samples were incubated overnight at 8 °C in the first antibody solution in a humid chamber.

The next day, a triple wash step with PBS + 1% albumin from calf serum (bovine serum albumin, BSA; PAA laboratories, Cölbe, Germany) was performed. This was followed by 2 h incubation with the fluorescent second antibodies (in PBS + 1% BSA): Alexa 488 FluorTM goat anti rabbit (1:1000; absorption: 488 nm; emission: 519 nm; Invitrogen, Karlsruhe, Germany) for LBP; Alexa FluorTM 568 goat anti mouse (1:1000; absorption: 478 nm; emission: 603 nm; Invitrogen) for TLR4. After washing twice with PBS and additionally once with Aqua bidest., the cover glasses or material samples were fixed on glass slides with the embedding medium ProLongGold (Invitrogen).

The antibodies that were used for this purpose can be found in Table 1.

The mRNA of the cell probes was extracted by chloroform (VWR, Langenfeld, Germany). After one centrifugation step, the upper phase, which contained the mRNA, was transferred to an RNAse-free tube with isopropanol (VWR). Followed by two centrifugation steps, the pellet was dissolved in 15 µL of RNAse-free water (Life Technologies, Darmstadt, Germany).

The primers for the RT-PCR analysis were all from Qiagen (Hilden, Germany) (see Table 2).

The fixation method was as follows:
SEM analysis: 2.5% glutaraldehyde (Carl Roth, Karlsruhe, Germany; in PBS) for 30 min;RT-PCR: no fixation; lysis of the cells by trizol (QIAzol, Qiagen, Hilden, Germany);Immunocytochemical marking:
○for LBP/TLR4 = 4% Paraformaldehyd solution (4% PFA in PBS) for 30 min;○for phalloidin/vinculin = 3.7% PFA (+10% Methanol; Merk Millipore, Schwalbach, Germany) + 0.2% Triton X100-solution for 30 min.



In order to analyse the significance of the results, an initial verification was carried out to determine whether the measured values could be allocated to a normal distribution (D’Agostino-Pearson omnibus normality test, Shapiro–Wilk normality test, Kolmogorov–Smirnov test with a Dallal–Wilkinson–Liliefor *p*-value; GraphPad Prism 6, GraphPad Software, La Jolla, San Diego, CA, USA). If a normal distribution was not present, the Kruskal–Wallis test (non-parametric test) or a *t*-test (Mann–Whitney Test) was used for more than two groups. The non-parametric *t*-test was used in cases in which a normal distribution of the data was present. It was assumed that the data in each group were distributed independently from each other. With the PCR tests, a pairwise fixed reallocation randomisation test was carried out using the REST program (Bloomingdale, IL, USA).

## 3. Results

### 3.1. Scanning Electron Microscope Images

The analysis of the untreated PEEK samples revealed that the surfaces had a characteristic grinding pattern (Figure 1A). Even after culture in NHDF and NHOst media, the indentations were still clearly visible (Figure 1B,C).

The titanium samples had a much rougher surface (Figure 1D). The structured surface relief was not changed by incubation in the two-growth media (Figure 1E,F).

In the control cultures, the osteoblasts on the coverslips (reference material) and the titanium samples presented a widely spread cytosoma (Figure 2A,C). In comparison, the osteoblasts (NHOst) on the PEEK surfaces had a more elongated orientation (red- and green-stained cells in Figure 2B). The orientation of the osteoblasts is probably attributable to the grinding pattern.

Under the influence of LPS, the NHOst formed numerous fine thin cytodendrites on the coverslips (Figure 2D). The osteoblasts also contracted on the PEEK and titanium samples to a greater extent than those grown on coverslips (Figure 2E,F).

A large number of fibroblasts (NHDF) grew and adhered to the surfaces of the coverslips and of the titanium samples. The cytosoma of the NHDF were mostly elongated (Figure 3A,C). The fibroblasts grew predominantly along the grinding profile on the PEEK surface (cells stained green and blue in Figure 3B).

LPS stimulation led to the fibroblasts having a more elongated orientation on the surface of the coverslips and PEEK samples (Figure 3D; red marked cells, Figure 3E). A large number of fibroblasts were spread extensively over the surface of the titanium (blue/yellow cells in Figure 3F).

### 3.2. Real-Time PCR Testing for Expression of LBP and TLR4

LPS incubation resulted in a strong increase of LBP expression (Factor 4.080 ± 0.44) in relation to that of the controls (mean C_T_ value of the controls: 25.37 ± 0.28; LPS: 23.44 ± 0.31; *n* = 8 samples; *p* = 0.001; Figure 4). A slight, but insignificant decrease (Factor 0.958 ± 0.067) of the TLR4 expression values was discernible between the control cultures and the LPS cultures (mean *C*_T_ value of the controls: 29.42 ± 0.18; LPS: 29.53 ± 0.10; *p* = 0.769; *n* = 8 samples; Figure 3).

A slight reduction of LBP expression (Factor 0.836 ± 0.09) was measured in the LPS-stimulated fibroblast cultures when compared with that of the controls (mean *C*_T_ value of the controls: 27.84 ± 0.23; LPS: 28.14 ± 0.16; *p*-value: 0.335; Figure 5). A slightly reduced amplification was also determined for TLR4 (Factor 0.788 ± 0.13) as a consequence of LPS incubation (mean *C*_T_ value of the controls: 30.99 ± 0.61; LPS: 31.41 ± 0.12; *p* = 0.785; Figure 4). There was a slight, but insignificant decrease in these values.

### 3.3. Evidence of Protein Expression of LBP and TLR4

A homogeneous distribution of LBP immunostaining was detected within the entire cytoplasm of the osteoblasts on all three tested materials (Figure 6A,D,G). With regard to TLR4, a similar intracellular distribution pattern was seen, although the perinuclear region fluoresced most intensively (Figure 6B,E,H; white arrow). A strong colocalisation of the LBP and TLR4 immune reactions was discernible on all materials (Figure 6C,H,I).

After LPS incubation, the greatest intensity of the antibody fluorescence against LBP and TLR4 was discernible in the perinuclear region of the cells grown on coverslips and the PEEK samples (Figure 7A,B,D,E; arrow). The elongated cytodendrites were also clearly immunostained on titanium (Figure 7H, arrowhead). The fluorescence intensity on all three of the materials was stronger than that under the corresponding control conditions.

In the fibroblasts, the distribution of LBP and TLR4 immunostaining in the cytoplasm was homogeneous (Figure 8A,B,D,E,G,H). The greatest intensity in all of the cultures was to be found in the perinuclear region (Figure 8A,B,D,E, arrows). A considerable colocalisation of LBP and TLR is discernible in the overlays of the images (Figure 8C,F,I).

LPS incubation triggered an increased LBP antibody reaction in the perinuclear region of the NHDF on all material samples (Figure 9A,D,G). When compared with the results under control conditions, the intensity of the TLR4 immune reaction was considerably reduced under LPS stimulation, with the only exception being in the cells grown on coverslips (Figure 9B,E,H).

### 3.4. Analysis of Actin Cytoskeleton and Cell Adhesion Contacts (Vinculin) in Osteoblasts and Fibroblasts

The fibrous bundles of the NHOst actin occurred in parallel lines or in diagonal directions in all the samples (Figure 10A,D,G). The cell boundaries and the perinuclear region displayed a strong anti-vinculin immune reaction on the surfaces of the coverslips and the titanium samples (Figure 10B,H). Detection of specific vinculin signals was unfortunately prevented by the strong autoflourescence of PEEK (Figure 10E). Considerable colocalisation was discernible between the actin cytoskeleton and the vinculin reaction (Figure 10C,I).

As in the control cultures, a parallel to diagonally oriented actin cytoskeleton was detected on all three of the used materials after LPS incubation (Figure 11A,D,G). A stronger punctate vinculin immunostaining was observed in the perinuclear region and in the cell extensions on the coverslips and titanium surfaces (Figure 11B,H; arrows). The PEEK samples again showed the great disadvantage of its strong autofluorescence (Figure 11E,F). Images of the cells grown on coverslips and titanium surfaces were superimposed and revealed vinculin signals in the area of actin-stained structures (Figure 11C,I).

The actin cytoskeleton of the fibroblasts was arranged in parallel lines (Figure 12A,D,G). The strongest intensity of the anti-vinculin reaction on the surfaces of the coverslips and titanium samples was found in the perinuclear region (Figure 12B,H; arrows). In this context, the location of the actin fibre bundles correlated with the homogeneous cytoplasmic vinculin distribution (Figure 12C,F,I).

The 24 h LPS incubation resulted in isolated gaps in the cellular syncytium of the fibroblasts (arrows in Figure 13A,D,G). The diagonal arrangement of the actin fibres was retained, as was the case in the control cultures.

A redistribution of the vinculin-immunostained adhesion contacts occurred from the perinuclear region to the periphery of the cytodendrites in cells grown on coverslips and titanium surfaces (Figure 13B,H). The vinculin immunostaining in cells grown on the PEEK surface showed a peripheral cytoplasmic distribution (Figure 13E). Notably, the vinculin immunostaining in the cells on the titanium samples was strongest when compared with that of the other materials (Figure 13H). The signals for the widely spread actin fibre bundles overlapped with the homogeneous vinculin distribution (Figure 13C,F,I).

## 4. Discussion

The scanning electron microscopy analyses revealed that the osteoblasts on the coverslips and titanium surfaces tended to be widely spread. Contrary to this, the NHOst on the PEEK samples had an elongated cell morphology.

One explanation for this result is that the relatively irregular surface of the coverslips, attributable to the PLL coating, encourages cell adhesion. The PEEK, on the other hand, has an extremely smooth surface, apart from the grinding pattern, making the adhesion of cells more difficult. Other authors have described poor cell growth on the surface of PEEK because of its inert chemical structure and hydrophobic surface [50,51]. Very smooth or very rough surfaces do not appear to benefit the growth of osteoblasts [52]. Though the roughness of the cover slips was not quantified, it was confirmed that the actin cytoskeleton of fibroblasts spread between fibrinogen spots on the glass surface [53].

All the material samples were sterilised using UV light before they were used. The irradiation (250–400 nm) results in the photocatalytic degeneration of the PEEK as a result of chain breakages, crossed reconnections, and the formation of carbonyl and hydroxyl groups. The formation of these functional groups may be the reason for the growth of the osteoblasts and fibroblasts on the PEEK.

One reason for the good adhesion of both cell types on titanium is that hydroxyl groups (–OH^−^) are formed on titanium dioxide surfaces in an aqueous environment. An apatite structure is formed in connection with free calcium ions (Ca^2+^) and phosphate groups (PO_4_^3−^) from the culture medium and activates the adhesion and growth of osteoblasts [54].

The fibroblasts on the coverslip surfaces and the PEEK surfaces were widely spread, whereas the NHDF on the titanium had an elongated morphology. In vitro studies have shown that PEEK supports the accumulation and proliferation of fibroblasts without having a negative effect on their cell growth [55,56]. Some authors have observed a high proliferation rate for epithelial cells and fibroblasts on smooth surfaces [57]. The accumulation of human epithelial cells and fibroblasts in the oral mucosa is promoted by a nano-rough surface [58]. Thus, osteoblasts prefer a rough surface, whereas fibroblasts grow better on smooth structures [59].

In this study, the material PEEK was shown to facilitate an osseointegration comparable to that of titanium. However, the SEM analysis confirms the working hypothesis that the inflammation reaction of the osteoblasts/fibroblast towards PEEK in the presence of LPS was weaker in comparison with that towards titanium under LPS incubation. This study did not allow an analysis of whether the surface structures of the materials and the LPS incubation had a later influence on the migration of the two cell types.

LPS has long been known to interact with the TLR4 receptor via the bond with LBP. Studies have detected the TLR4 receptor in the cell membrane of osteoblasts [60]. LBP is used as a biomarker for the diagnosis of local bacterial infections, as it is the primary inflammation marker for LPS in the serum [61].

With regard to cDNA expression of LBP, a considerably greater increase was measured in the NHOst cultures that had been stimulated with LPS when compared with the corresponding controls. However, the TLR4 expression in the osteoblasts was not altered under treatment with LPS.

Considerable immunostaining for both LBP and TLR4 was also detected on all tested material surfaces within the entire cytoplasm of the osteoblasts under control conditions. LPS stimulation spurred an intensification of the immune reaction to the LBP and TLR4 proteins. The more intense immunostaining in the perinuclear region is an indication of increased protein expression.

The LPS incubation of fibroblasts resulted in slight reductions in their LBP and TLR4 cDNA concentrations, although these reductions were not significant. A study conducted on the human gingiva revealed that a 10 µg/mL LPS incubation led to the largest increase in the mRNA concentrations for LBP and TLR4 [62].

With regard to fibroblasts under control conditions, the considerable focal density of the LBP and TLR4 signals in the perinuclear regions was conspicuous, indicating the existence of strong protein synthesis in this area. The TLR4 receptor participates in the proliferation of skin fibroblasts [63]. The fibroblasts appeared to react less sensitively to the LPS incubation, presumably reflecting their task as connective tissue cells.

Vinculin is a focal adhesion protein. Increased vinculin expression is connected to increased adhesion strength [64]. This is of relevance for the formation of cell–cell contacts and the transference of mechanical loads to the cell membrane. It could be expected that only the vinculin antibody marks the focal adhesion points where the extracellular matrix is connected with the intracellular actin cytoskeleton, as shown in Figure 11 (showing osteoblasts on the coverslips after LPS incubation). In the other figures and on the other materials, the perinuclear region shows the highest intensity of fluorescence for the osteoblasts and fibroblasts. One explanation for this phenomenon could be that both cell types first build up some stabile contacts with the material surface in the region of the cell soma. The materials “coverslip” and “PEEK” have a relatively smooth surface. Both cell types must spread out to get in contact with the material to provide a higher vinculin protein expression. This may be the reason for the high fluorescence intensity in the perinuclear region. Additionally, filopodia move around to build up new adhesion contact points with the materials’ surfaces. Actin fibers in the filopodia support this by analyzing the surrounding surface [65]. Especially in strong migration cells like fibroblasts, long cell extensions can be found [66].

After the LPS incubation, isolated gaps in the cellular syncytium of the fibroblasts could be found on all materials. A possible explanation could be that the fibroblasts built up less adhesion contacts with the materials and were detached by the phalloidin and vinculin immune reaction. One hint for this was the parallel orientation of the actin fibers as they were in the control culture. Thus, a reorganisation of the actin cytoskeleton was not possible. It is possible that the LPS incubation led to a higher detachment of the cell extensions and the cells had contact almost exlusively with the material surface in the perinuclear region. This is why only this part shows a strong vinculin immune reaction.

By changing the shape of their actin filaments, osteoblasts were able to adapt their cytoskeleton to the geometry of the surface of the substrate [67]. The actin fibres of MG-63 osteosarcoma cells on titanium samples are arranged in long and organised stress fibres that extend throughout the cytosoma and terminate at the surface structures [68]. Under LPS stimulation, the osteoblasts contracted in an elongated manner on all the used materials.

The fibroblasts, in particular, had a parallel, slightly crossing actin fibre system discernible in cells grown on the smooth surfaces of the coverslips and the PEEK samples. The extensive diagonal course of the actin bundles on the titanium samples indicates that the cells attempted to come into close contact with the rough surface.

LPS stimulation caused isolated gaps to occur in the cell layers of the fibroblast cultures. However, the parallel arrangement of the actin fibres persisted, as in the control cultures. Contrary to this, as far as human fibroblasts of the gingiva are concerned, a reorganisation of the actin filaments was observed when the cells were subjected to LPS and LBP treatment (100 ng/mL) for 24 h [69]. The strongest signals for the ECM protein vinculin were observed in the cytosoma after LPS treatment. This is an additional indication that LPS is responsible for an increased detachment of the cytodendrites.

To the best of our knowledge, the present study provides evidence that PEEK represents an alternative to titanium as an implant material, as in the case of zirconium. The reasons are the comparable cellular adhesion of the osteoblasts on PEEK, as on titanium, and the diminished inflammation reaction of the osteoblasts and fibroblasts. To provoke osseointegration, cellular adhesion is indispensable. The aim of this study was not to denigrate titanium, but rather to indicate that PEEK may serve as a future alternative material to titanium. Clinical studies are necessary to verify this conclusion, as there are only in vitro results to be discussed at the moment. No scientific data currently exist with regard to the intraossary clinical use of PEEK, despite the fact that the benefits (colour, e-module, hypo-allergenicity) when compared with titanium have previously been favourably discussed [70,71].

Authors should discuss these results and how they can be interpreted from the perspective of previous studies and of working hypotheses. The findings and their implications should be discussed in the broadest possible context. Future research directions may also be highlighted.

## 5. Conclusions

PEEK enables the adhesion of human osteoblasts and fibroblasts under inflammatory environmental conditions that are similar to those of titanium.PEEK supports inflammatory processes to a lesser extent when compared with titanium.

## Figures and Tables

**Figure 1 materials-12-02739-f001:**
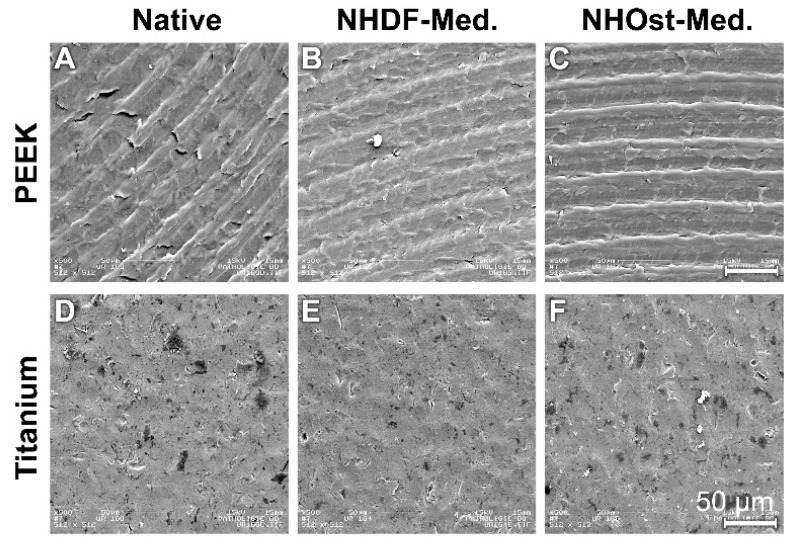
Material probes without cells. The scale unit in each picture is the same. (**A**) PEEK Native; (**B**) PEEK NHDF-Med.; (**C**) PEEK NHOst-Med.; (**D**) Titanium Native; (**E**) Titanium NHDF-Med.; (**F**) Titanium NHOst-Med.

**Figure 2 materials-12-02739-f002:**
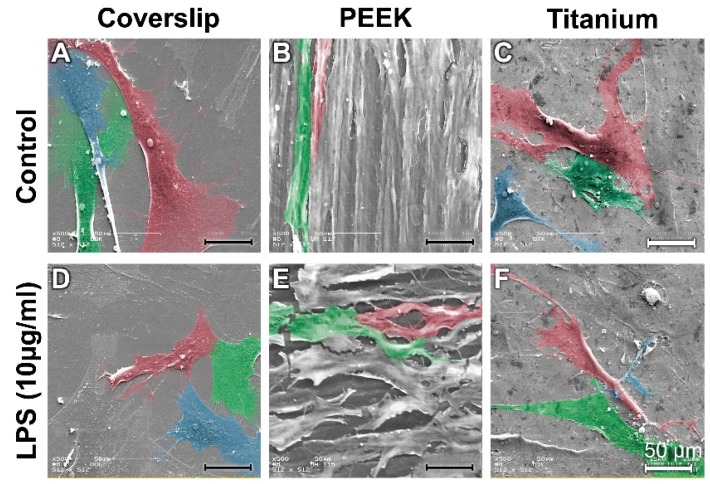
Scanning electron microscopy (SEM) images of osteoblasts on coverslips on polyether ether ketone (PEEK) and on titanium samples in the proliferation medium and after lipopolysaccharide (LPS) incubation. (**A**) Control Coverslip; (**B**) Control PEEK; (**C**) Control Titanium; (**D**) LPS Titanium; (**E**) LPS PEEk; (**F**) LPS Titanium.

**Figure 3 materials-12-02739-f003:**
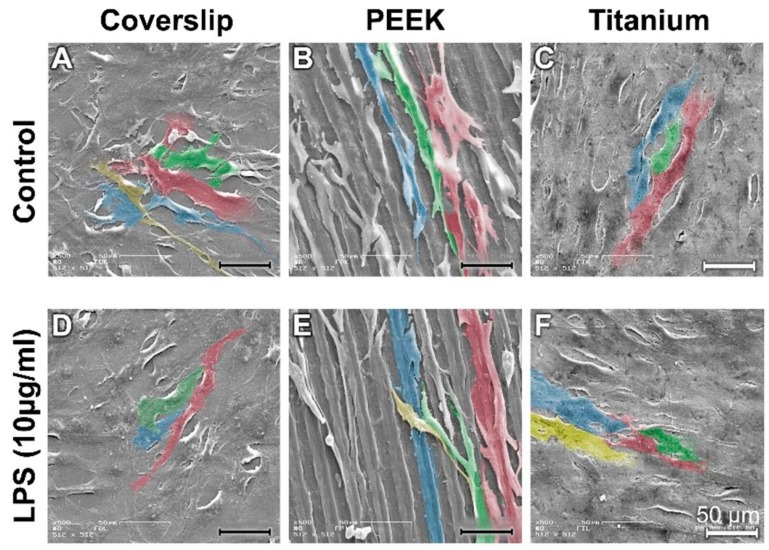
SEM images of fibroblasts on coverslips on PEEK and on titanium samples under control conditions and after LPS incubation. (**A**) Control Coverslip; (**B**) Control PEEK; (**C**) Control Titanium; (**D**) LPS Coverslip; (**E**) LPS PEEK; (**F**) LPS Titanium.

**Figure 4 materials-12-02739-f004:**
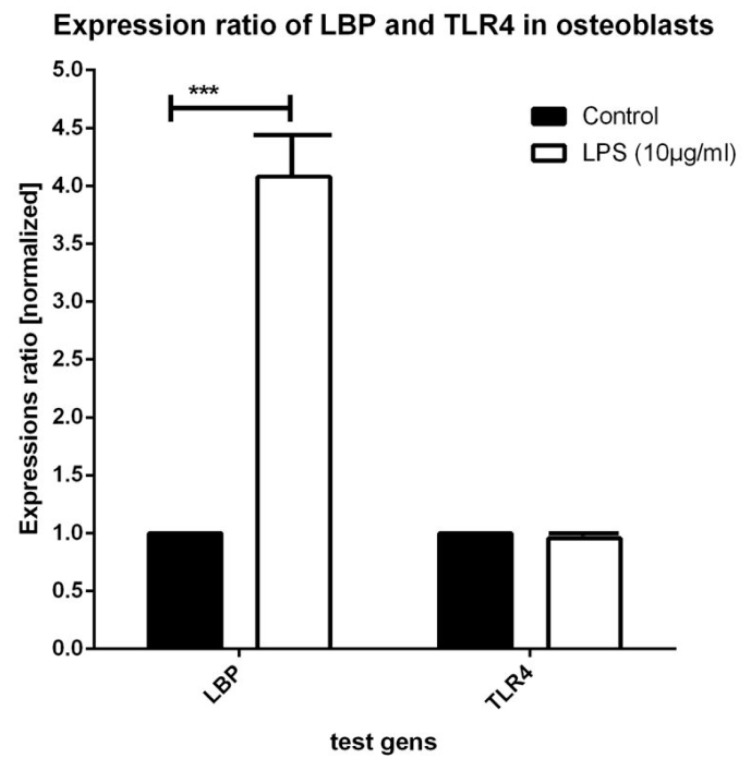
Expression of LPS-binding protein (LBP) and of toll-like receptor 4 (TLR4) in osteoblasts under control and LPS culture conditions (10 µg/mL, 24 h).

**Figure 5 materials-12-02739-f005:**
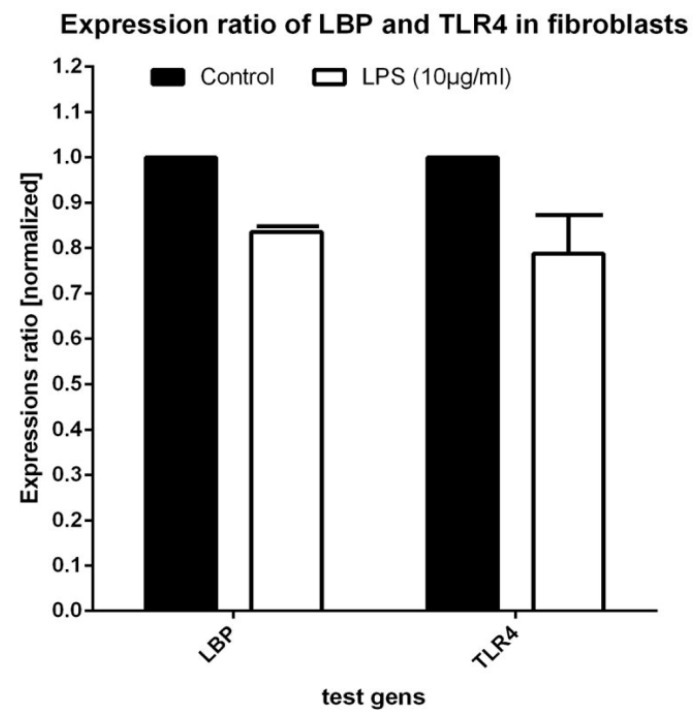
Expression of LBP and TLR4 in fibroblasts under control and LPS culture conditions (10 µg/mL, 24 h).

**Figure 6 materials-12-02739-f006:**
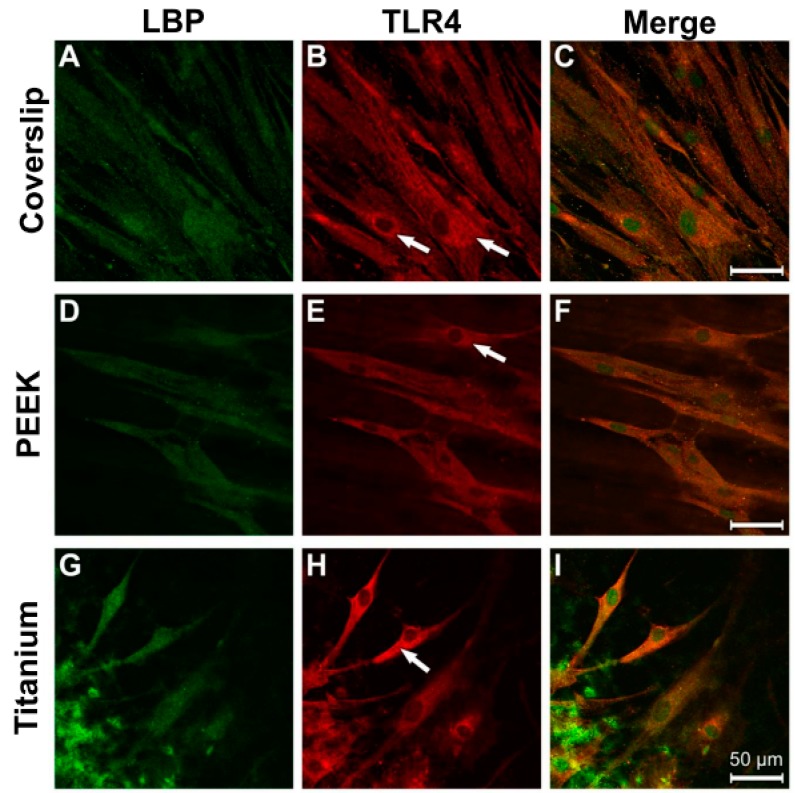
Immunocytochemical evidence for LBP and TLR4 in osteoblasts under physiological culture conditions. (**A**) Coverslip LPB; (**B**) Coverslip TLR4; (**C**) Coverslip Merge; (**D**) PEEK LBP; (**E**) PEEK TLR4; (**F**) PEEK Merge; (**G**) Titanium LBP; (**H**) Titanium TLR4; (**I**) Titanium Merge.

**Figure 7 materials-12-02739-f007:**
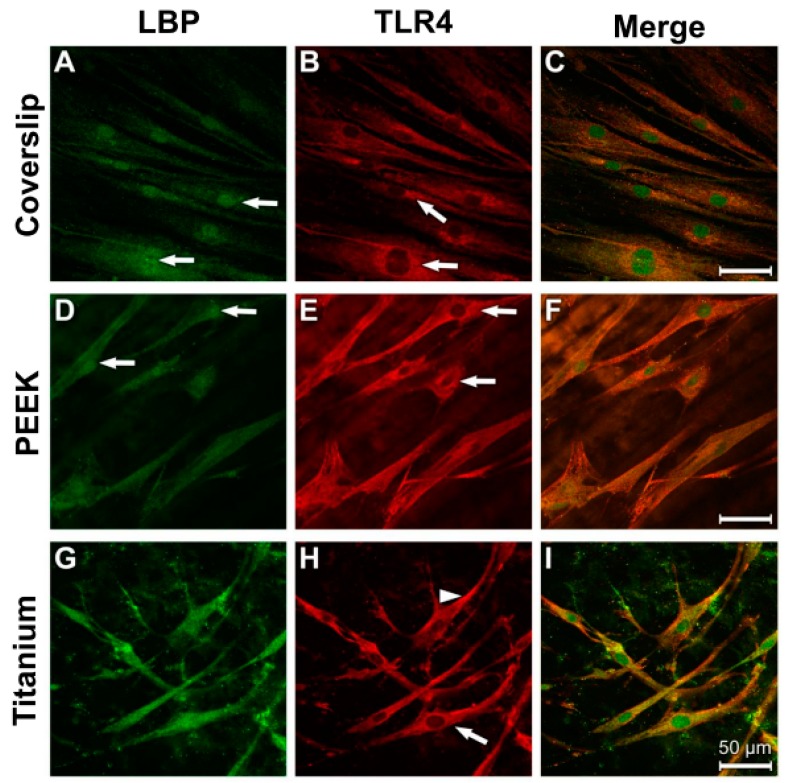
Antibody reaction for LBP and TRL4 in human osteoblasts under LPS stimulation (10 µg/mL, 24 h). (**A**) Coverslip LBP; (**B**) Coverslip TLR4; (**C**) Coverslip Merge; (**D**) PEEK LBP; (**E**) PEEK TLR4; (**F**) PEEK Merge; (**G**) Titanium LBP; (**H**) Titanium TLR4; (**I**) Titanium Merge.

**Figure 8 materials-12-02739-f008:**
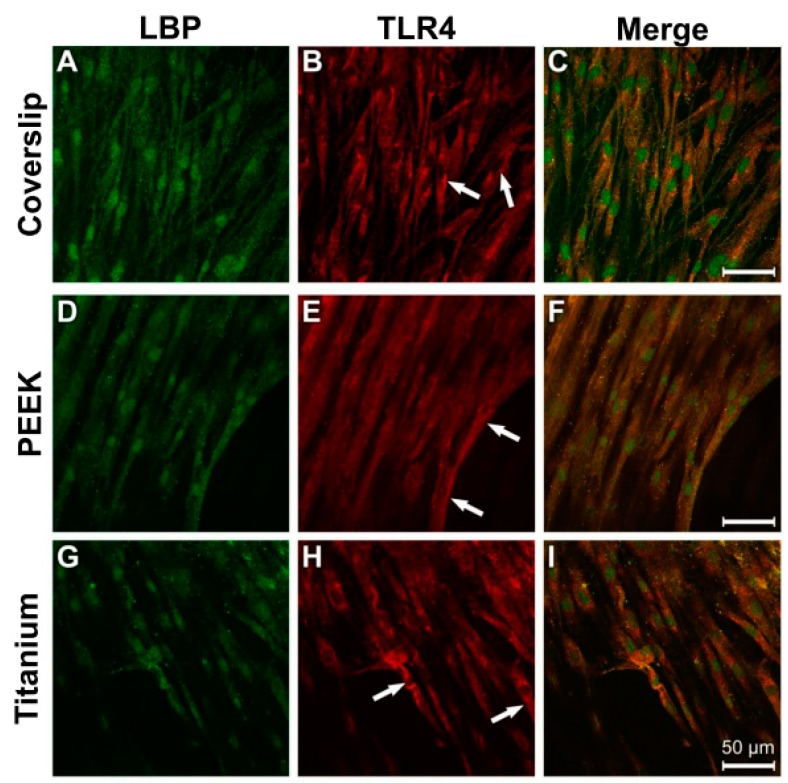
Immunocytochemical evidence of LBP and TLR4 in fibroblasts under physiological culture conditions. (**A**) Coverslip LBP; (**B**) Coverslip TLR4; (**C**) Coverslip Merge; (**D**) PEEK LBP; (**E**) PEEK TLR4; (**F**)PEEK Merge; (**G**) Titanium LBP; (**H**) Titanium TLR4; (**I**) Titanium Merge.

**Figure 9 materials-12-02739-f009:**
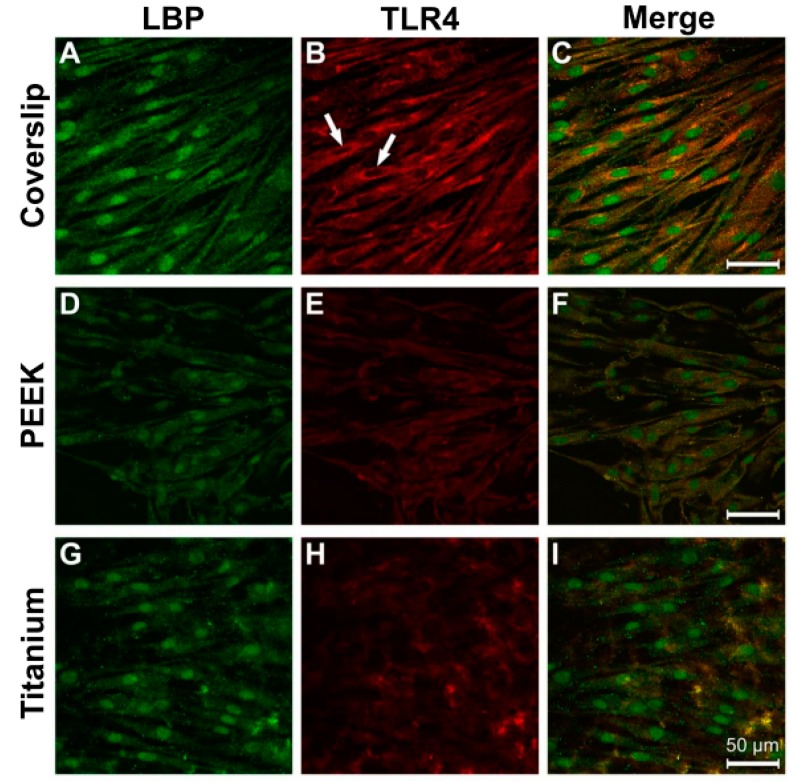
Antibody reaction to LBP and TRL4 in human fibroblasts under LPS stimulation (10 µg/mL, 24 h). (**A**) Coverslip LBP; (**B**) Coverslip TLR4; (**C**) Coverslip Merge; (**D**) PEEK LPB; (**E**) PEEK TLR4; (**F**) PEEK Merge; (**G**) Titanium LBP; (**H**) Titanium TLR4; (**I**) Titanium Merge.

**Figure 10 materials-12-02739-f010:**
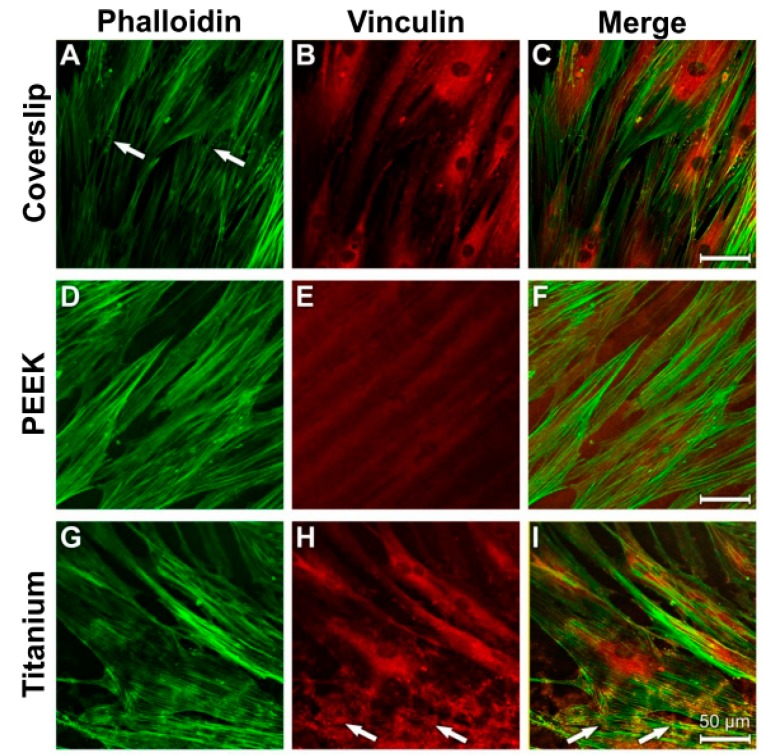
Staining of actin cytoskeleton and cell adhesion contacts in osteoblasts under control culture conditions. (**A**) Coverslip Phalloidin; (**B**) Coverslip Vinculin; (**C**) Coverslip Merge; (**D**) PEEK Phalloidin; (**E**) PEEK Vinculin; (**F**) PEEK Merge; (**G**) Titanium Phalloidin; (**H**) Titanium Vinculin; (**I**) Titanium Merge.

**Figure 11 materials-12-02739-f011:**
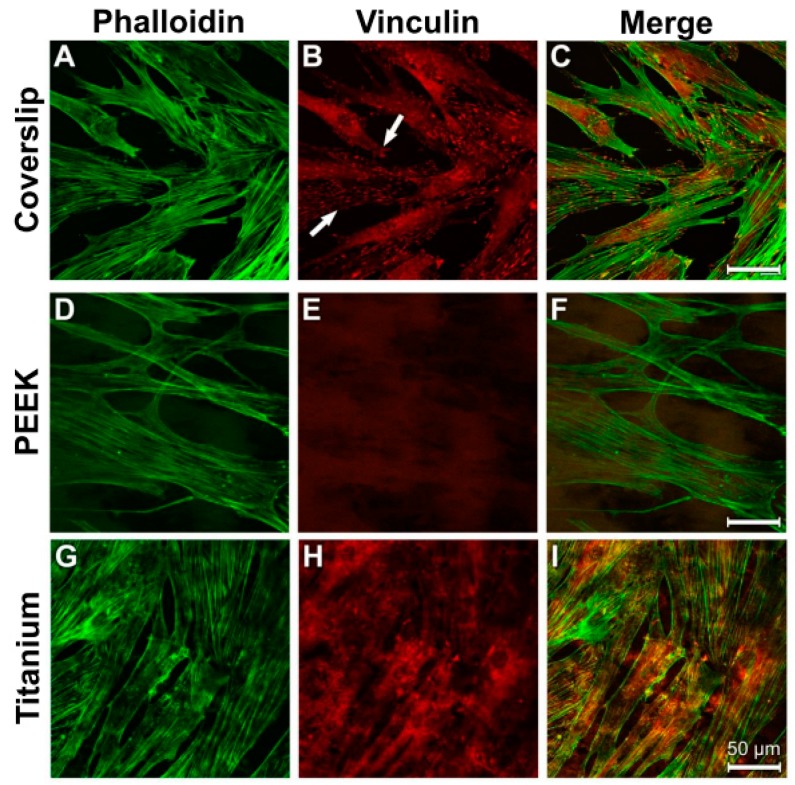
Staining of actin cytoskeleton and cell-material contacts in osteoblasts under induced inflammatory conditions. (**A**) Coverslip Phalloidin; (**B**) Coverslip Vinculin; (**C**) Coverslip Merge; (**D**) PEEK Phalloidin; (**E**) PEEK Vinculin; (**F**) PEEK Merge; (**G**) Titanium Phalloidin; (**H**) Titanium Vinculin; (**I**) Titanium Merge.

**Figure 12 materials-12-02739-f012:**
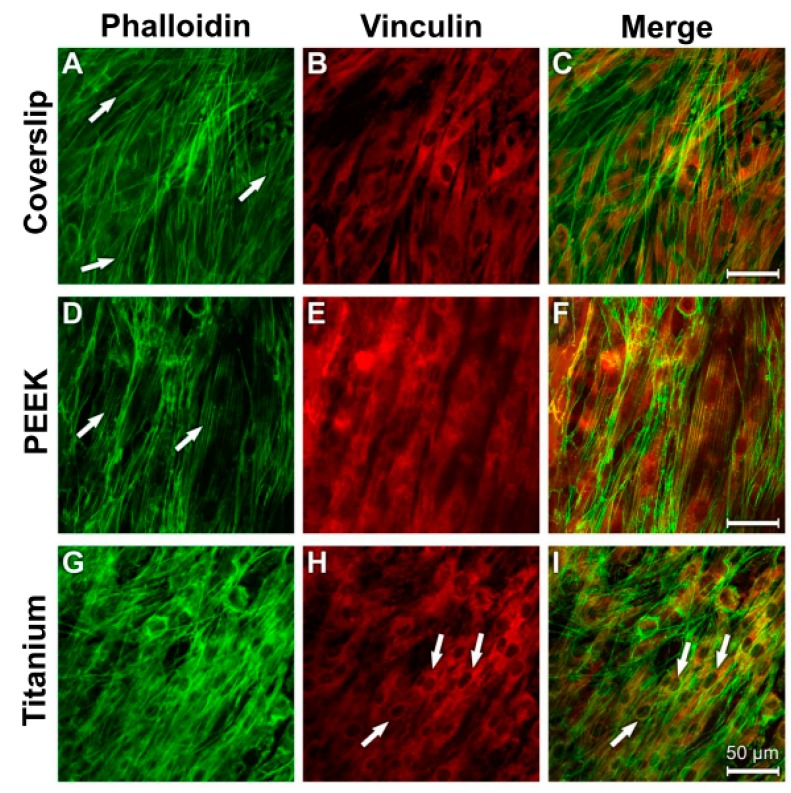
Staining of actin cytoskeleton and cell adhesion contacts in human fibroblasts under control culture conditions. (**A**) Coverslip Phalloidin; (**B**) Coverslip Vinculin; (**C**) Coverslip Merge; (**D**) PEEK Phalloidin; (**E**) PEEK Vinculin; (**F**) PEEK Merge; (**G**) Titanium Merge; (**H**) Titanium Vinculin; (**I**) Titanium Merge.

**Figure 13 materials-12-02739-f013:**
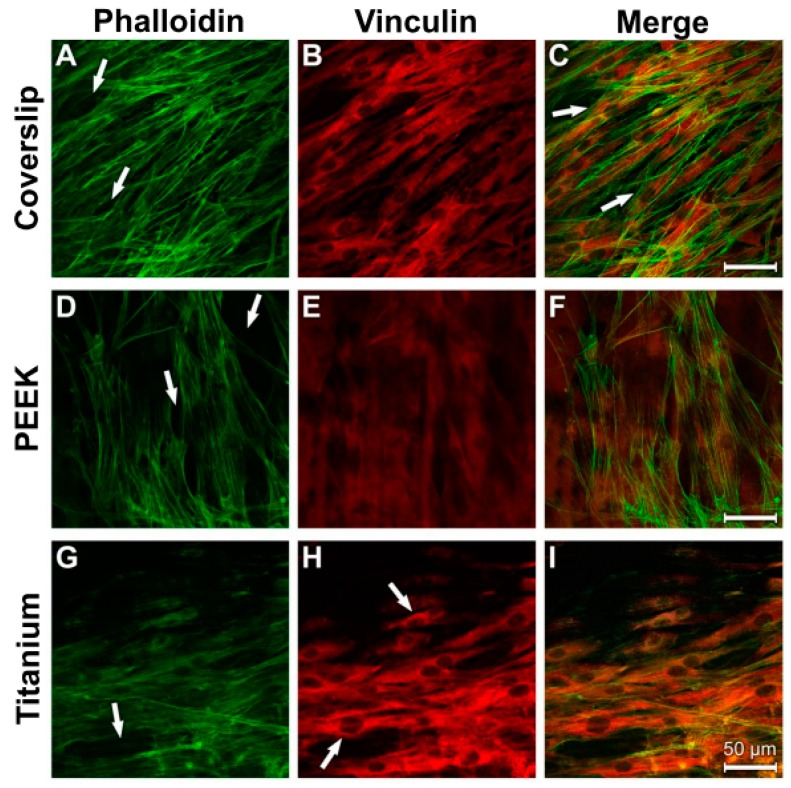
Staining of actin cytoskeleton and cell adhesion contacts in human fibroblasts after LPS incubation. (**A**) Coverslip Phalloidin; (**B**) Coverslip Vinculin; (**C**) Coverslip Merge; (**D**) PEEK Phalloidin; (**E**) PEEK Vinculin; (**F**) PEEK Merge; (**G**) Titanium Phalloidin; (**H**) Titanium Vinculin; (**I**) Titanium Merge.

**Table 1 materials-12-02739-t001:** Antibodies used for immunocytochemistry. LBP, lipopolysaccharide (LPS)-binding protein; TLR4, toll-like receptor 4.

Name	Company	Dilution Ratio
Alexa goat 488 anti-mouse	Invitrogen (Karlsruhe, Germany)	1/1000
Alexa goat 488 anti-rabbit	Invitrogen (Karlsruhe, Germany)	1/1000
Alexa goat 568 anti-mouse	Invitrogen (Karlsruhe, Germany)	1/1000
rabbit-anti-human LBP (PA5-21642)	Thermo Scientific (Watham, MA, USA)	1/75
Phalloidin Atto-488	Sigma-Aldrich (Taufkirchen, Germany)	1/1000
Maus-anti-human TLR4 (76B357.1, (ab22048))	Abcam (Cambridge, UK)	1/75
Monoclonal Anti-Vinculin (clone hVIN-1)	Sigma-Aldrich (Taufkirchen, Germany)	1/500

**Table 2 materials-12-02739-t002:** The primer used for LBP, TLR4, and the normalisation genes.

Primer	Function	Gen (Length)	Primer Length [bp]	Catalogue-No.
Hs_LBP_1_SG	lipopolysaccharide-binding protein	NM_004139 (1894 bp)	79	QT00027293
Hs_TLR4_1_SG	Toll-like receptor 4	NM_003266 (5781 bp)	102	QT00035238
Hs_ACTB_1_SG	β-Actin	NM_001101 (1852 bp)	146	QT00095431
Hs_CYC1_1_SG	Cytochrome C1	NM_001916 (1251 bp)	123	QT00209454
Hs_EIF4A2_1_SG	Eukaryotic initiation faktor 4A2	NM_001967 (1905 bp)	87	QT00079226
Hs_GAPDH_2_SG	Glycerinaldehyd-3-phosphat-dehydrogenase	NM_002046 (1421 bp)	119	QT01192646
Hs_HMBS_1_SG	Hydroxymethylbilane Synthase	NM_000190 (1526 bp)	107	QT00014462
Hs_RRN18s_1_SG	Ribosomal 18s RNA	X03205 (1869 bp)	149	QT00199367

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
