# Peer review of "Adhesion Behaviour of Primary Human Osteoblasts and Fibroblasts on Polyether Ether Ketone Compared with Titanium under In Vitro Lipopolysaccharide Incubation"

_materials, 2019, doi:10.3390/ma12172739_

Round 1
Reviewer 1 Report
In ‘Adhesion Behaviour of Primary Human Osteoblasts and Fibroblasts on Polyether Ether Ketone compared with Titanium under Induced Inflammation – a Pilot Study’, Benz et al. analyzed the adhesion of osteoblasts and fibroblasts on PEEK and titanium in the presence of LPS.
The main concern about this manuscript is that it seems like a number of the figures are incorrect. They do not match the captions or the other text in the section. For the figures with fluorescence microscopy images, it is also impossible to tell if these are of osteoblasts or fibroblasts from the images, so it raises some doubts if these figures are also correct or are mislabeled. It has been very difficult to provide a complete review because of these errors; nevertheless, some specific comments are below.
The term in the title ‘induced inflammation – a pilot study’ is somewhat unclear. It would be perhaps more precise to change this to in vitro lipopolysaccharide incubation.
Lines 65-67: Some more information about LPS-binding protein and its relevance should be included in the introduction.
In the materials and methods section, it would be helpful to break it up into subsections with headings.
Lines 83-91: What was the seeding density of the cells on the different samples? Were the cells seeded first in normal medium to attach and then was it changed to the LPS containing medium or was LPS present also during initial cell attachment?
Line 102: What primers specifically were used?
Line 109: How was the immunocytochemical staining performed? Were negative controls performed to confirm staining specificity? Also, what secondary antibody was used with the antvinculin antibody?
Line 125: What is the composition of the proliferation medium?
Figures 1 and 2: Is it possible to also include SEM images of the surfaces without cells? Do the surfaces have the same topography? I am wondering why the cells are adopting an elongated and seemingly oriented morphology on the PEEK samples.
Line 142: How/why are the red marked cells considered more elongated in Figure 2E? Was image analysis performed? What criteria were used for this distinction?
Figures 3 and 4 appear to be incorrect. The captions suggest that these should show expression of LBP and TLR4; however, the graphs are labeled with viability and formazan absorption.
Lines 158-162: The p-values of 0.335 and 0.785 suggest that there is no difference between the LPS and control groups, so it seems incorrect to claim a slight reduction/amplification of LBP and TLR4.
Figures 5 and 6 also appear to be incorrect and showing the PCR data instead of the immunocytochemistry.
Lines 177-178 and 189: What data support the conclusion that the intensity of the staining was increased/reduced? Was quantification performed? Were multiple and independent samples analyzed?
Figures 9 and 10 appear to be incorrect, as the figures are labeled with LBP and TLR4 while the caption talks about actin cytoskeletol and cell adhesion contacts.
Lines 198-199: Why wasn’t a proper fluorophore selected to avoid overlap with the autofluorescence?
Line 217: Why would vinculin be homogeneously distributed in the cytoplasm?
Line 237: The roughness coming from the grinding pattern should be quantified.
Line 241: The sterilization procedure should be described in the methods.
Lines 258-259: What is meant by the inflammation reaction is weaker? In this study, the effect of LPS on cell behavior is being examined. How is this specifically linked to inflammation?
Lines 313-315, 320-321, 337-339: This seems to be left from the template.
Lines 320-335: No information on author contributions, funding, or acknowledgments are provided.
Author Response
Answers to reviewer 1
Comments and Suggestions for Authors
In ‘Adhesion Behaviour of Primary Human Osteoblasts and Fibroblasts on Polyether Ether Ketone compared with Titanium under Induced Inflammation – a Pilot Study’, Benz et al. analyzed the adhesion of osteoblasts and fibroblasts on PEEK and titanium in the presence of LPS.
The main concern about this manuscript is that it seems like a number of the figures are incorrect. They do not match the captions or the other text in the section. For the figures with fluorescence microscopy images, it is also impossible to tell if these are of osteoblasts or fibroblasts from the images, so it raises some doubts if these figures are also correct or are mislabeled.
Thank you very much for this important point, the figures were not in the correct order. We deeply apologize for this mistake. We changed the order of the figures.
It has been very difficult to provide a complete review because of these errors; nevertheless, some specific comments are below.
The term in the title ‘induced inflammation – a pilot study’ is somewhat unclear. It would be perhaps more precise to change this to in vitro lipopolysaccharide incubation.
Thank you for your note. We changed the title.
Lines 65-67: Some more information about LPS-binding protein and its relevance should be included in the introduction.
We added more information about LPS-binding protein and underlined its relevance by citing scientific literature.
In the materials and methods section, it would be helpful to break it up into subsections with headings.
Thank you for this point, we underlined the different methods to present it more clearly.
Lines 83-91: What was the seeding density of the cells on the different samples?
This information was added to the manuscript in lines 135/136.
Were the cells seeded first in normal medium to attach and then was it changed to the LPS containing medium or was LPS present also during initial cell attachment?
The cells were first seeded on the different materials in the specific growth medium of the NHDF/NHOst of the companies PromoCell/LONZA. After four days of growth, the medium was changed to growth medium without/+ 10µg LPS for 24h. This information is already mentioned in the m&m section.
Line 102: What primers specifically were used?
Primers that were used are now mentioned in table 2.
Line 109: How was the immunocytochemical staining performed?
An exact description can now be found in lines 159-177.
Were negative controls performed to confirm staining specificity?
Negative controls (without first antibody) were performed, but are not shown to shorten the manuscript.
Also, what secondary antibody was used with the antvinculin antibody?
The secondary antibody Alexa FluorTM 568 goat-anti-mouse (1:1000; absorption: 478nm; emission: 603nm; Invitrogen, Karlsruhe, Germany) was used with the antivinculin antibody and is now mentioned in the manuscript.
Line 125: What is the composition of the proliferation medium?
Only the specific growth mediums of the companies PromoCell/LONZA were used for the cultivation of the NHDF/NHOst. The exact compositions of these mediums are not mentioned.
Figures 1 and 2: Is it possible to also include SEM images of the surfaces without cells? Do the surfaces have the same topography?
Please have a look at figure 1 now.
I am wondering why the cells are adopting an elongated and seemingly oriented morphology on the PEEK samples.
Our explanation for this phenomenon is that the company MEDICON (eG, Tuttlingen) cut the probes with a rotary method. This is the reason for the characteristic ring pattern on the surface of the PEEK probes. The NHDF and NHOst use the surface topography for their cell orientation and growth.
Line 142: How/why are the red marked cells considered more elongated in Figure 2E? Was image analysis performed? What criteria were used for this distinction?
One explanation for the more elongated morphology of the red marked cell in figure 2E is the characteristic ring pattern on the surface of the PEEK probes. The cells migrated into the impressions and tried to get in contact with the material due to the relative smooth surface.
Figures 3 and 4 appear to be incorrect. The captions suggest that these should show expression of LBP and TLR4; however, the graphs are labeled with viability and formazan absorption.
In the primary version of the manuscript we had more figures. By uploading the figures to the system we used the order chosen in the first manuscript and didn’t change that. We apologize for this mistake.
Lines 158-162: The p-values of 0.335 and 0.785 suggest that there is no difference between the LPS and control groups, so it seems incorrect to claim a slight reduction/amplification of LBP and TLR4.
The significance of the RT-PCR data was analysized by a Pair wise fixed reallocation randomisation test with the REST-programm (Bloomingdale, USA). The result of this analysis was, that there is a slight reduction of LBP and TLR4, despite the high p-values. This is shown in the figure 4.
Figures 5 and 6 also appear to be incorrect and showing the PCR data instead of the immunocytochemistry.
See mentioned above to figure 3 and 4.
Lines 177-178 and 189: What data support the conclusion that the intensity of the staining was increased/reduced? Was quantification performed? Were multiple and independent samples analyzed?
We tried to verify the immune staining by a quantification method. But there were to many interferences of the fluorescence, especially of the material PEEK, to get significant results. So the conclusion, that the intensity of the staining was increased/reduced, is our subjective view.
We analysed at least three independent samples for each immune staining.
Figures 9 and 10 appear to be incorrect, as the figures are labeled with LBP and TLR4 while the caption talks about actin cytoskeletol and cell adhesion contacts.
See mentioned above in figure 3 and 4.
Lines 198-199: Why wasn’t a proper fluorophore selected to avoid overlap with the autofluorescence?
In the pilot test we had least auto-fluorescence of the material PEEK. Later on, we had some problems with our fluorescence microscope but could use another one in another department. They have other fluorescence filters and this could be the explanation for the high auto-fluorescence of PEEK in the figures.
Line 217: Why would vinculin be homogeneously distributed in the cytoplasm?
The anti-vinculin reaction of the fibroblasts on the surfaces of the coverslips and titanium samples was more or less homogeneously distributed in the cytoplasm but the strongest intensity was found in the perinuclear region. Another possible explanation for the homogeneously distribution is that the NHDF try to get in contact with the smooth PEEK surface by multiple adhesions contact points that are linked by vinculin with the intracellular actin cytoskeleton.
Line 237: The roughness coming from the grinding pattern should be quantified.
The roughness of the cover slips, coming from the grinding pattern with PLL, was not to be quantified. But another study confirms, that the actin cytoskelett of fibroblast is spread out between fibrinogen spots on the glass surface. We added this information in lines 329-331.
Line 241: The sterilization procedure should be described in the methods.
The sterilization procedure is now being described in the methods in lines 134-138.
Lines 258-259: What is meant by the inflammation reaction is weaker? In this study, the effect of LPS on cell behavior is being examined. How is this specifically linked to inflammation?
LPS incubation of cells is often using a model system for an inflammation reaction of the tissue. It is a preferable model for the simulation of bacterial infection in in-vitro studies. Incubation with living bacteria in cell culture could be dangerous because of contamination of other cells.
Lines 313-315, 320-321, 337-339: This seems to be left from the template.
Lines 320-335: No information on author contributions, funding, or acknowledgments are provided.
All informations are now provided.
Reviewer 2 Report
In this manuscript, the authors present and defend PEEK as a valid alternative to Titanium, specifically when a pro-inflammatory microenvironment is presented, and highlight the capacity of supporting inflammatory processes of this polymer in vitro.
The data are solid, and can support the results. The manuscript is of great interest to the researchers in the field of implanting materials.
The manuscript could be considered for possible publication if the authors could address the following comments/doubts:
-Has the titanium surface some kind of specific treatment that can alter the surface microtopography and consequently cell deposition attachment (e.g. sand-blasted, acid-etched)?
-If the manuscript addresses the cell adhesion onto these surfaces in a inflammatory environment, it would be interesting performing some studies using immune cell lines (monocytes, macrophages...). The authors should add a paragraph addressing this issue.
-The adhesion of cells onto a determined surface depends greatly onto the protein layer formed onto the surface after implantation. The protein depostion patterns is dependent on multiple variables, among which the surface chemistry, microtopography, etc. This might define the adhesion behaviour of this cells. The authors should state a comment regarding the influence of protein deposition onto the cell attachement.
-Authors should add the primer's sequences to the manuscript, as well as the mRNA extraction method.
-Authors should state the fixation method.
-Scale bar units on the images should be more readable. Suggestion: you don't need to repeat the scale units on every picture.
Author Response
Answers to reviewer 2
Comments and Suggestions for Authors
In this manuscript, the authors present and defend PEEK as a valid alternative to Titanium, specifically when a pro-inflammatory microenvironment is presented, and highlight the capacity of supporting inflammatory processes of this polymer in vitro.
The data are solid, and can support the results. The manuscript is of great interest to the researchers in the field of implanting materials.
The manuscript could be considered for possible publication if the authors could address the following comments/doubts:
-Has the titanium surface some kind of specific treatment that can alter the surface microtopography and consequently cell deposition attachment (e.g. sand-blasted, acid-etched)?
On the REM images the surface was very “rough” but the samples came directly from the producer. Specifications were: Titanium alloy: 3.7165 acc. to DIN ISO 5832-3 and ASTM F 136.
We don’t have any information about a specific pre-treatment. If you wish we could include an image of the REM.
After receipt the material samples (titanium and PEEK) were
o cleaned/degreased in 70% ethanol for 6 hours (room temp, RT) on a shaking plate,
o dried in the sterile bench
o wrapped in aluminium foil,
o autoclaved (121°C, 1bar) for 20 min
o stored at RT
o sterilized with UV light for 15 min before sowing the cells.
We thought this information is not very useful for the reader. If you think we should add this to our manuscript, please let us know. Thank you!
-If the manuscript addresses the cell adhesion onto these surfaces in a inflammatory environment, it would be interesting performing some studies using immune cell lines (monocytes, macrophages...). The authors should add a paragraph addressing this issue.
We added a paragraph addressing this issue.
-The adhesion of cells onto a determined surface depends greatly onto the protein layer formed onto the surface after implantation. The protein depostion patterns is dependent on multiple variables, among which the surface chemistry, microtopography, etc. This might define the adhesion behaviour of this cells. The authors should state a comment regarding the influence of protein deposition onto the cell attachement.
We stated a comment regarding the influence of protein deposition and stated the signifance by literature research.
-Authors should add the primer's sequences to the manuscript, as well as the mRNA extraction method.
Primer’s sequences as well as the mRNA extraction method were included into the manuscript.
-Authors should state the fixation method.
The fixation method is now stated in the Materials&Methods section.
-Scale bar units on the images should be more readable. Suggestion: you don't need to repeat the scale units on every picture.
Thank you for your note. We made the scale unit bigger in the last picture of the figure 1 and 2.
Round 2
Reviewer 1 Report
The authors have addressed a number of concerns in their revision of this manuscript. However, I still have the following comments:
1. The title has not been changed in online journal system.
2. Line 19: Are actin and vinculin extracellular matrix binding proteins? Isn’t actin fully intracellular?
3. The materials and methods section should be divided up into sections with headings (2.1, 2.2, etc.).
4. Figure 1: For the part that is unreadable, this can be cropped off and the larger scale bars kept.
5. Lines 223-226: It would be helpful to mention that the orientation of the osteoblasts is probably due to the grinding pattern.
6. Figures 4 and 5: What material was used as the substrate in these experiments to measure the gene expression levels of LBP and TLR4?
7. Lines 254-258: Do you mean to say that there is a slight but insignificant decrease in these values? It would be helpful to write this in the text.
8. Lines 297-298: Where is the colocalization of the actin and vinculin?
9. Line 348: Do you mean potassium ions or calcium ion? (incorrect abbreviation)
10. Lines 383-385: More explanation is needed regarding what would be expected from the vinculin staining (that it is staining the focal adhesions) in comparison with the observed results, that the vinculin appears to be throughout the cytoplasm and in the perinuclear region. This could indicate nonspecific staining.
11. Line 357: It seems to be a bit of a stretch to claim osseointegration when no in vivo studies were performed.
12. The manuscript should be carefully edited for grammar and typos.
Author Response
Comments and Suggestions for Authors
The authors have addressed a number of concerns in their revision of this manuscript. However, I still have the following comments:
The title has not been changed in online journal system.The title has now been changed in the online journal system.
Line 19: Are actin and vinculin extracellular matrix binding proteins? Isn’t actin fully intracellular?This is correct, actin isn´t an extracellular matrix binding protein. Actin in cells is mostly concentrated in the cytoplasm and to a smaller part also in the nucleus (Guharoy et al., 2013). The actin cytoskeleton in the cytoplasm consists mostly of long actin fibres that cross the cytoplasm.
Vinculin connects the integrins of the extracellular matrix with the intracellular actin cytoskeleton (Carisey and Ballestrem, 2011, Nabavi et al., 2011, Pivodova et al., 2011). These contact points are defined as focal adhesion points (FA) of the cells with the growing surface (Biggs et al., 2008). The adhesion points controlled the adhesion and migration of osteoblasts (Carisey and Ballestrem, 2011, Ciobanasu et al., 2013).
The materials and methods section should be divided up into sections with headings (2.1, 2.2, etc.).The sections are now headed by subheadings.
Figure 1: For the part that is unreadable, this can be cropped off and the larger scale bars kept.We kept the part that is unreadable to show that the figures are original pictures and not changed except the coloring of the cells for the better evaluation of the cell morphology. In the current version we largened it to provide a better reading.
Lines 223-226: It would be helpful to mention that the orientation of the osteoblasts is probably due to the grinding pattern.Thank you for your hint. For the PEEK probes it is clear, that the orientation of the osteoblasts is due to the grinding pattern. The flat cell morphology of the osteoblasts on titanium is probably due to the rough surface of the material. The cells try to spread out to get in contact with the surface. Also osteoclasts need the contact to their neighbors cells to grow. This is a second reason for the more or less parallel orientation of the NHOst.
The coverslips were coated with the protein lysine before seeding the cells. Probably, the coating layer isn´t homogenously distributed over the surface. Perhaps it is more orientated in fine lines and the osteoblasts use these lines for their cell growing.
We mentioned yout hint in line 229.
Figures 4 and 5: What material was used as the substrate in these experiments to measure the gene expression levels of LBP and TLR4?
For the gene expression experiments of LBP and TLR4, the osteoblasts and fibroblasts were seeded on coverslips (coated with poly-L-lysine). In preliminary studies we found, that the RNA concentration after growing on titanium and PEEK probes were too less for the RT-PCR amplification.
Lines 254-258: Do you mean to say that there is a slight but insignificant decrease in these values? It would be helpful to write this in the text.
This is correct! There is a slight but insignificant decrease in the TLR4 expression of the osteoblastsbetween the control cultures and the LPS cultures. We added this information in the text.
Lines 297-298: Where is the colocalization of the actin and vinculin?
In the corresponding figure 7, we found a colocalization of the actin and vinculin signals in the osteoblasts. Especially in the perinuclear region a high intensity of the antibody fluorescence against LBP and TLR4 was discernible.We focus in our description of the figure 7 on the higher fluorescence intensity in comparison with the corresponding control conditions.
Line 348: Do you mean potassium ions or calcium ion? (incorrect abbreviation)
Sorry for the mistake. The abbreviation is correct, so we changed it to calcium ion.
Lines 383-385: More explanation is needed regarding what would be expected from the vinculin staining (that it is staining the focal adhesions) in comparison with the observed results, that the vinculin appears to be throughout the cytoplasm and in the perinuclear region. This could indicate nonspecific staining.
Thank you for your hint. We added som more information about this in lines 419-438.
Line 357: It seems to be a bit of a stretch to claim osseointegration when no in vivo studies were performed.
The examination was designed as a pilot study for an up-following in-vivoanalysis. The osseointegration behavior of materials in most cases is first tested in in-vitrostudies.
We rewrote this sentence: In this study, it was shown that the material PEEK facilitates an osseointegration comparable with that of titanium. (see line 390,391)
The manuscript should be carefully edited for grammar and typos.
The manuscript has been proof-read by a professional proof-reading service.
BIGGS, M. J., RICHARDS, R. G., MCFARLANE, S., WILKINSON, C. D., OREFFO, R. O. & DALBY, M. J. 2008. Adhesion formation of primary human osteoblasts and the functional response of mesenchymal stem cells to 330nm deep microgrooves. J R Soc Interface,5,1231-42.
CARISEY, A. & BALLESTREM, C. 2011. Vinculin, an adapter protein in control of cell adhesion signalling. Eur J Cell Biol,90,157-63.
CIOBANASU, C., FAIVRE, B. & LE CLAINCHE, C. 2013. Integrating actin dynamics, mechanotransduction and integrin activation: the multiple functions of actin binding proteins in focal adhesions. Eur J Cell Biol,92,339-48.
GUHAROY, M., SZABO, B., CONTRERAS MARTOS, S., KOSOL, S. & TOMPA, P. 2013. Intrinsic structural disorder in cytoskeletal proteins. Cytoskeleton (Hoboken),70,550-71.
NABAVI, N., KHANDANI, A., CAMIRAND, A. & HARRISON, R. E. 2011. Effects of microgravity on osteoclast bone resorption and osteoblast cytoskeletal organization and adhesion. Bone,49,965-74.
PIVODOVA, V., FRANKOVA, J. & ULRICHOVA, J. 2011. Osteoblast and gingival fibroblast markers in dental implant studies. Biomed Pap Med Fac Univ Palacky Olomouc Czech Repub,155,109-16.
Reviewer 2 Report
I thank the authors for addressing the questions on my first review.
Nonetheless, there are still some questions/comments pending after this second review with the alterations the authors have made to the manuscript, before being accepted for publication.
-At a general level, the English employed to write this manuscript should be revised and improved. I have found some errors regarding the verbal times employed to write some sentences (lines 93-96 are too confusing, should be rewritten), and some gramatical errors (e.g. "fist physiological tissue"; "Glutardialdehyd"; "collagen fibbers", "24er well plate" among others...)
-Materials and methods should be re-organized and divided into sections to make the reading more clear (example:putting the fixation method for each technique altogether makes reading too confusing; you can re-organize into 2.1 SEM 2.2 RT-PCR 2.3 Immunostaining...)
-The authors should give more emphasis to adherence proteins like Vitronectin and Fibronectin, as the scope of the manuscript is based on adhesion behavior, instead of giving so much importance to albumin, in my opinion.
Author Response
Comments and Suggestions for Authors
I thank the authors for addressing the questions on my first review.
Nonetheless, there are still some questions/comments pending after this second review with the alterations the authors have made to the manuscript, before being accepted for publication.
-At a general level, the English employed to write this manuscript should be revised and improved. I have found some errors regarding the verbal times employed to write some sentences (lines 93-96 are too confusing, should be rewritten), and some gramatical errors (e.g. "fist physiological tissue"; "Glutardialdehyd"; "collagen fibbers", "24er well plate" among others...)
The manuscript has been proof-read by a professional proof-reading service.
-Materials and methods should be re-organized and divided into sections to make the reading more clear (example:putting the fixation method for each technique altogether makes reading too confusing; you can re-organize into 2.1 SEM 2.2 RT-PCR 2.3 Immunostaining...)
This has been edited.
-The authors should give more emphasis to adherence proteins like Vitronectin and Fibronectin, as the scope of the manuscript is based on adhesion behavior, instead of giving so much importance to albumin, in my opinion.
We gave more emphasis to this important part. Please see lines 112-142.